# Understanding the menstrual health self-care practices and experiences among women with physical disabilities in rural Nepal: A qualitative study

**Suyasha Adhikari**[1]*, **Pabitra Neupane**[1,2], **Anisha Shrestha**[2,3], **Manasi Sharma**[4], **Karishma Bhandari**[5], **Amit Timilsina**[2]

**1** Master of Arts in Gender Studies, Tribhuvan University, Kathmandu, Bagmati, Nepal, **2** Research and Community Development Center, Kathmandu, Nepal, **3** School of Health Science, University of Toledo, Toledo, Ohio, United States of America, **4** Bachelor in Public Health, Manmohan Memorial Institute of Health Sciences, Tribhuvan University, Kathmandu, Bagmati, Nepal, **5** Bachelor in Public Health, Central Department of Public Health, Tribhuvan University, Kathmandu, Bagmati, Nepal

☯ These authors contributed equally to this paper.
* suyashaadhikari321@gmail.com

## Abstract

### Background

Women with disabilities often face significant challenges in promoting and practicing menstrual health, which compromises their quality of life and overall well-being. Limited studies have been conducted on women with physical disabilities and their self-care practices related to menstrual health in rural Nepal. Thus, this study aims to explore the differential experience and self-care practices for menstrual health among married women with physical disabilities in Karnali Province of Nepal.

### Methods

The study used an exploratory qualitative method utilizing purposive sampling. Data collection was conducted through in-depth interviews and focus group discussions, in which 31 married women with disabilities participated. A thematic analysis approach was employed, in which codes were generated inductively and organized into sub-themes, then consolidated into themes, which were subsequently presented as the findings supported by excerpts.

### Result

The majority of the participants relied on gut feelings, and bodily symptoms such as head itches, mood swings, an increase in body temperature, home remedies for period cramps, and self-examination of the colour, odour, and consistency of menstrual blood for self-management. Participants self-tested for pregnancy in case of

**Data availability statement:** All relevant data are within the paper and its Supporting Information files.

**Funding:** The author(s) received no specific funding for this work.

**Competing interests:** The authors have declared that no competing interests exist.

missed period while self-monitoring their blood pressure and menstrual blood for health diagnosis. The participants had misconceptions such as passing on disability to their fetus due to pain-management medication, practiced seclusion, avoided consumption of dairy products, fruits, and foods, and had limited knowledge, access, and misinformation regarding menstrual products beyond cotton clothes and menstrual pads, which suggests low self-awareness regarding menstrual health management.

## Conclusion

Access to disability-inclusive menstrual health information through optimal utilization of digital platforms and community sensitization events, knowledge and access to diverse menstrual product choices, mobilization of community volunteers for social and behaviour change initiatives, and the self-care inclusive health policies promoting social protection and menstrual health self-care provisions are key for decreased sexual and reproductive morbidity, and improved quality of life among women living with disability.

---

## Introduction

Self-care has been an important tool and action that complements the health system for promoting quality, affordable, efficient, accessible, and client-centric health services [1]. WHO defines self-care as the knowledge and capacity of individuals, families, and communities to take action that promotes and maintains good health, well-being, prevents and manages disease, infection, illness, or disability independently or with the assistance of a health professional [2]. Its framework identifies self-awareness, self-management, and self-testing as key components of the self-care approach in health [2]. Self-awareness refers to the individual's capacity for self-help, self-education, self-regulation, self-efficacy, and self-determination. Self-management involves practices such as self-medication, self-treatment, self-examination, self-injection, self-administration, and self-use of health services or products, and self-testing includes activities like self-sampling, self-screening, self-diagnosis, self-collection, and self-monitoring [3]. This approach promotes universal health by increasing choice and autonomy when it is accessible, acceptable, and affordable, and also supports greater self-determination, self-efficacy, and engagement in health for both self-carers and caregivers [3].

The self-care in Sexual and Reproductive Health (SRH) is less studied and prioritized, particularly in rural settings of low and middle-income countries like Nepal [4]. This neglect is largely due to deeply rooted taboos, myths, and misconceptions surrounding SRH [5]. Menstrual health is a vital component of SRH, as both are shaped by common biological processes and influenced by similar social determinants of health. [6]. In Nepal, menstruation continues to be shrouded in silence, stigma, and harmful cultural beliefs, often treated as an "untouchable" or "silent" issue [7].

Menstrual health refers to overall physical, mental, and social well-being concerning the menstrual cycle, encompassing more than just the absence of illness. It encompasses access to accurate and age-appropriate information, the ability to manage menstruation hygienically using affordable products and supportive facilities, and timely medical care for any menstrual-related concerns [8–10].It also involves a stigma-free environment that promotes informed decision-making, and the right to participate fully in all aspects of life without facing discrimination, exclusion, or violence due to menstruation [11]. Over 1.9 billion people have menstruation worldwide, and the World Bank estimates that 500 million people do not have menstrual health and are denied the freedom to control their monthly cycle respectably and healthily [12].

According to the Nepal Census 2021 A.D., 2.2% of the country's population has some form of disability [13]. Among them, approximately 2.5% of men and 2.0% of women have some form of disability [13]. Among ten types of disabilities: Physical disability, disability related to vision, disability related to hearing, deaf-blind, disability related to voice and speech, psycho-social disability, intellectual disability, disability associated with haemophilia, disability associated with autism, and multiple disabilities [14].

Physical disability is the most common, accounting for 36.75% of all reported disabilities [13]. It is a limitation that arises in the operation of physical parts, use, and movement in a person due to problems in nerves, muscles, and composition, and the operation of activities of bones, and a person whose height is excessively lower than the average height according to age [14]. Furthermore, disability is also categorised based on its severity: profound (persons who face difficulty to perform day-to day activities even with continuous support of others), severe disability (persons who requires others support continuously to perform personal activities and involve in social activities), moderate disability (persons who can regularly participate in daily and social activities if physical, environmental, educational barrier is ended and mild disability (persons who can regularly participate in daily and social activities if physical and environmental barriers is dismantled [14].

According to the 2021 Census, Karnali Province, though it comprises 10 districts, has the lowest population among the seven provinces of Nepal, with 1,688,412 people, representing 5.79% of the country's total population [13,15]. However, it has the highest prevalence of persons with disabilities, at 3.1%, which is slightly above the national average. Of those with disabilities, 55.6% are male and 44.4% are female, with physical disabilities being the most common form [13].

In order to facilitate inclusive development, the government of Nepal has provided disability identity cards and a monthly allowance for profound and severe disabilities under social security allowances [14]. The initiatives, such as launching national awareness campaigns, free sanitary pad distribution in 2019 to school girls in public schools across all seven provinces, improving water, sanitation, and hygiene (WASH) facilities in schools in collaboration with CSOs, criminalization of menstrual seclusion practice *Chhaupadi pratha,* and introducing gender studies to aware on gender related issues to common from 2009, are aiding efforts to eliminate malpractices during menstruation in Nepal and in Karnali province [16–18]. However, due to social norms surrounding menstruation, as well as the stigma associated with disability, women with disabilities frequently face multifaceted stigma in managing their menstrual hygiene with dignity [19,20]. In rural Nepal, the overall situation of menstruation is worse in comparison to the other parts of the country [21]. The health of women with disabilities is impacted by unhygienic menstruation behaviours such as inadequate support to use menstrual products, inaccessible toilet facilities, disposal facilities, overlooking signs of infections, etc., making them more susceptible to Reproductive Tract Infections, Pelvic Inflammatory Diseases, and other reproductive health issues [22–24]. Thus, prioritizing self-care can help empower women with disabilities to manage and prioritize their optimum health and well-being [1].

Self-care is particularly important for women with physical disabilities to access sexual and reproductive health services, such as menstrual hygiene, where stigma might otherwise deter them from seeking necessary care [4,25]. The importance of self-care practices during menstruation for women with physical disabilities becomes even more pronounced when coupled with the centralized health system, limited resources, infrastructure, and geographical challenges [4,24,25]. However, there are limited studies that aim to explore self-care menstrual hygiene practices among women with

physical disabilities to reduce reproductive morbidity, such as reproductive tract infection, and promote overall health and well-being [22,26,27]. Considering the marginalization due to geography, disability status, and gender, this paper aims to understand the differential experience of women living with physical disabilities, explore existing self-care practices on menstrual hygiene, and uncover underlying socio-ecological factors that influence menstrual health self-care in the Karnali province of Nepal.

## Methods and materials

### Study design and study site

The exploratory qualitative study method using a *conceptual framework for self-care intervention* in the study design and throughout the study to understand the differential experience of menstrual health management and to explore self-care practices during menstruation among women with physical disabilities [28]. The study was conducted in the Surkhet and Mugu districts of Karnali Province, Nepal.

### Participants and sampling

Purposive Sampling was conducted to identify and select the participants. A list of target groups was created, prioritizing those who were willing to participate in the study and could clearly describe their differential experiences in line with the objective of the study. The list was prepared with the help of three organizations working for persons with disabilities: Disable Rehabilitation & Rural Development Organization (DARRDO) Nepal, Blind Women Association Nepal (BWAN) Surkhet Chapter, and Blind Youth Association Nepal (BYAN) Surkhet Chapter. These organizations helped to identify eligible participants with physical disabilities. From the obtained list, married women aged 18–49 living with physical disabilities (moderate and severe physical disability) were selected. Only individuals who voluntarily agreed to participate and provided informed consent to share their experiences were listed. To meet the objective of the study, women with physical disabilities under 18 years of age and women with mild disability, and participants who did not provide consent for the interview were excluded. Based on these criteria, the principal investigator finalized the list of participants.

The participants in the study included individuals with varying degrees of severity of disability, categorized based on severe and moderate disabilities. Among the 31 participants, three were between 18 and 20 years of age, including two with moderate disabilities and one with a severe disability. Four participants fell within the 21–25 age group, with three having moderate disabilities and one having a severe disability. Eight participants were in the 26–30 age group; five had moderate disabilities, and three had severe disabilities. The largest group was aged 31–35, comprising eleven participants, eight with moderate disabilities and three with severe disabilities. Additionally, five participants were above 35 years old, of whom three had moderate disabilities and two had severe disabilities. In total, 21 participants had moderate disabilities and 10 had severe disabilities.

### Data collection tools and procedures

The data was collected from 1st September 2023–5th October 2023. To ensure proper ethical guidelines, written ethical approval was obtained from the Ethical Review Board of Nepal Health Research Council (Ref. Number 157) before the data collection process. The researchers developed a semi-structured in-depth interview (IDI) guide and focus group discussion (FGD) guide in the Nepali language through an iterative process, ensuring it effectively addressed relevant topics that are suitable for the target population and local context. A pre-test was conducted among 5 participants to assess the clarity, sensitivity, applicability, and face validity of the tool. Based on the result of the pre-test, necessary revisions were made and the final version of the semi-structured interview guides.

Following the finalization of the semi-interview guides, 21 IDIs and 1 FGD involving 10 participants, consisting total of 31 participants was conducted for the data collection by four trained female enumerators. IDI was conducted to collect

detailed personalized experiences and contextually relevant insights from individual participants without any external influence, while one FGD was conducted to triangulate and validate the findings from IDI, and gather a broader range of opinions, shared beliefs, collective attitudes, and group dynamics among participants, helping to reach data saturation.

Before initiating interviews, the enumerators read aloud the information and consent forms to the participants. Considering the physical disabilities of participants, verbal consent was taken before the interview and during the interview by recording on the recorder. All interviews and focus group discussions were conducted in the Nepali language. The participants were oriented about the study title and objective, and that they could leave the question or leave the interview whenever they wanted without any explanation. The data was collected at the residence of participants in a separate room where the enumerator and participant were only present. Adequate follow-up was made by the first and second author with the enumerators, field notes were taken, and a de-briefing of the interview was done to ensure trustworthiness of the data.

On average, each IDI lasted approximately 35 minutes, while the FGD session lasted around one hour. Data collection was stopped upon reaching data saturation, the point at which no new themes or significant insights emerged from additional interviews or discussions, affirming that the sample size was sufficient for comprehensive analysis while avoiding redundancy. The framework of data saturation for this study has been adapted from Fush.et. al. Are we there yet? [29].

### Data processing and analysis

Since the IDIs and a FGD were conducted in Nepali, the first stage of data processing involved verbatim transcribing audio recordings in Nepali. These transcripts were then translated verbatim into the English language. The transcriptions and translations of the qualitative data were done manually. To ensure accuracy and consistency, the translated transcripts were carefully reviewed for translation quality. Dedoose software was used to support the data management, particularly for the inductive coding [30].

Data were analysed using thematic analysis as guided by Braun and Clarke [31]. This method involves identifying, organizing, and interpreting recurring patterns within qualitative data, referred to as themes. The analysis followed six systematic steps: becoming familiar with the data, generating initial codes, identifying potential sub-themes and themes, reviewing and refining sub-themes and themes, defining and naming themes, and finally, compiling the analysis into a report, interpreting the themes and sub-themes supported by the excerpts. [32]. The data was analysed using an inductive thematic analysis approach, allowing themes and concepts to emerge directly from the data. This approach was chosen to ensure that the analysis emerged from the perspectives and experiences of the participants, rather than imposing pre-defined categories or frameworks.

**Ethical consideration.** Prior to the conduct of the study, ethical approval was obtained from the Nepal Health Research Council (Reference number: 157). The research followed Helsinki ethical guidance and biomedical ethical principles (autonomy, beneficence, non-maleficence, and justice.) throughout the study [33,34]. To strictly maintain anonymity and confidentiality, all the collected data was pseudonymized using unique participant IDs. Personal identifying information was removed during the transcription and translation process.

### Result

In total, 45 codes, two themes (*Existing self-care menstrual practices* and *Facilitators and barriers for adoption of menstrual self-care practices*), and five subthemes (*Self-management, Self-testing, Self-awareness, Facilitators for self-care practices,* and G*aps in self-care adoption for menstrual practices*) were developed as stated in Table 1.

### Existing self-care menstrual practices

**Self-management of menstrual practices.** This result section presents how women with physical disabilities manage their menstrual cycles, hygiene, health, and sanitation, encompassing overall menstrual hygiene management. It details

Table 1. Codebook to showcase codes, sub-categories, and categories.

| Themes | Sub-themes | Codes |
|---|---|---|
| Existing self-care menstrual practices | Self-management | Use of Ayurvedic and herbal medicines; Use of hot bags; Use of herbs and homemade remedies; Self-massage; Light food consumption; Intake of more fluids; Approaching health services in case of severe pain; Food avoidance; Examining the colour of blood; Flow of blood; Tracking the date of menstruation; Self-administration of medicines as suggested; Choice and use of pads; Disposal of pads; Stress and pain management |
| | Self-testing | Use of pregnancy kit in the absence of menstruation; self-evaluation of consistency and colour of blood; Use of calendars for menstrual cycle self-monitoring; Trust the physical body response |
| | Self-Awareness | Awareness regarding the menstrual cycle; Information regarding menstrual hygiene management; Awareness regarding menstrual products; Awareness regarding information source; Prioritization regarding menstrual health management; Availability of information; Awareness regarding intergenerational practice; Awareness regarding home remedies; Help-seeking behaviour |
| Facilitators and barriers for adoption of menstrual self-care practices | Facilitators for self-care in menstrual practices | Knowledge and skills to fight against the existing taboo; Participating in workshops and training; Information through health institutions; Information through peers and family; Use of digital online platforms; Use of social media; Textbook; Information through magazines and newspapers; Information through radio and TV; Family and partner support; Good behaviour by healthcare professionals; Empowerment of women with physical disabilities |
| | Existing gap for the adoption of menstrual self-care practice | Existing gender roles, Lack of menstrual health-focused policy at the workplace, Limited prioritization of menstrual hygiene management, Limited importance of self-care in menstrual practices, and Limited understanding regarding the importance of self-care |

practices such as self-medication, self-treatment, self-examination, and self-use that these women adopt to manage their menstrual hygiene.

In this study, the majority of the participants were found to use homemade remedies more frequently in comparison to allopathic medicines such as Meftal, Ibuprofen, Paracetamol, among others, to ease and treat menstrual cramps. It was found that they had ingrained beliefs about harming their bodies, which can also result in disability in their children if medicines are taken. Therefore, homemade remedies such as tea with ghee, warm honey water, plain lukewarm water, and self-message were preferred practices to alleviate menstrual cramps and pain.

*No, I do not take any pharmaceutical medicines to alleviate my menstrual cramps as I have heard they are not good for our body in the long run; therefore, I ask my mother to make me warm honey water to reduce menstrual cramps during my period. -* Woman with moderate physical disability.

Likewise, some of the participants without limbs stated the difficulties of self-medication; as a result, they relied on their husbands or family members for medication to reduce menstrual cramps. So, they were dependent upon their husbands or family members for any medication to relieve menstrual cramps.

*Since I do not have arms, I cannot do anything; rather, I ask my husband or my mother-in-law to provide me with luke-warm water during my period to reduce pain, and they provide it too. -* Woman with severe physical disability.

The use of menstrual products such as homemade pads, commercially produced pads, and menstrual underwear among women with disabilities varied according to their awareness level, availability, accessibility, and affordability. In rural settings, the majority of the women with physical disabilities were found to use clothes and torn pieces of saree, while in urban areas, most of them used synthetic menstrual pads. Likewise, none of them were found to use menstrual cups.

*I use cotton clothes and torn pieces saree as a menstrual product because I can't afford sanitary pads every month, though it is said to be safe and hygienic. -* Woman with severe disability.

While the majority of the participants were aware of both synthetic and homemade pads, only a few were familiar with menstrual underwear. Participants from urban areas preferred menstrual underwear due to its convenience, ease of use, and reduced need for frequent changes compared to synthetic pads. However, upon further probing, some participants reported not preferring menstrual underwear despite their availability, citing difficulties with usage and challenges in maintaining hygiene.

*Even though other women like me (women with physical disabilities) might feel convenient while using period panties but I do not feel the same because of my leg problem; it feels difficult for me to sit down and wash that panties for my next use, so instead I use synthetic pads. -* Woman with moderate physical disability.

One of the participants also emphasized the importance of hygiene over comfort, highlighting the potential health risks of unhygienic menstrual products of the health harm of unhygienic menstrual practices, and the importance of hygiene practice over comfort.

*No, I do not prefer menstrual underwear because they take a long time to dry, especially in winter, and because of their damp nature, yeasts will grow. Therefore, I either use cotton clothes or sanitary pads, upon their availability. -* Woman with severe physical disability.

No significant differences were found in the use of menstrual products based on the severity of disability. Most women with both moderate and severe physical disabilities living in urban areas preferred synthetic pads, although this varied for those who were financially unstable, with very few going for menstrual underwear. In contrast, women in rural areas often used torn pieces of saree and cloth due to limited resources and geographical barriers for going to buy menstrual products.

*My one leg is short, and I have no problem using pads as I put a support system in my leg; however, I have heard menstrual underwear could have been more comfortable, but again, it is expensive and difficult to wash. -* Woman with moderate physical disability.

Similarly, the majority of women with disabilities who acquired their disability later in life faced more challenges using pads compared to those who had a disability from early childhood.

*It was easier for me to use pads before an accident; however, after losing one arm, I always find it challenging to use pads. Even it has been one year of the accident, I take help from my husband to put on pads on my underwear. -* Woman with severe disability.

The disposal of homemade as well as synthetic pads varied according to geography and degree of disability. In rural areas, the majority of the participants stated throwing pads in the jungle, river, or toilets, and a few of them in dustbins, whereas in urban areas, many of the women were found to practice disposal of pads in dustbins. For women with severe physical disabilities, it was difficult to dispose of pads properly, even though in the dustbins.

*I can't bend my body properly, so I rely on my daughter to help wrap the used pads in paper and dispose of them. -* Woman with severe physical disability.

Upon further probing, it was found that those who practiced the proper disposal of pads had acquired this knowledge from sources such as school community-based organization (CBO) programs, or social media like Facebook and TikTok.

*I dispose of pads in the dustbin at school. At home, I wrap the pad in plastic and dispose of it in sac bags. Later, dumped into the vehicle that collects waste from houses. -* Woman with moderate disability.

In addition, female community health volunteers (FCHVs) played a pivotal role in disseminating knowledge and information about menstrual hygiene management, enabling women with disabilities to take care of themselves during menstruation. This included managing back and stomach pain, using and disposing of pads properly, and taking necessary medications.

*My experience regarding menstrual hygiene management after attending the training by our FCHV ma'am was far better than before. I knew the proper disposal of used pads. Before, I was too shy to dry my cloth pads in the sun, but now I don't feel shy at all. -* Woman with moderate physical disability.

A few women with disabilities also mentioned receiving information on menstruation and menstrual hygiene from traditional birth attendants.

*Sudeni mother (midwife) used to teach us to keep ourselves clean during menstruation to maintain hygiene. She has taught us that if we do not keep ourselves clean during menstruation, our bodies will smell bad as we bleed, and we might also get infections that potentially harm our uterus. Therefore, I keep myself clean during menstruation. -* Woman with moderate disability.

**Self-testing of menstrual practices.** Self-testing of menstruation refers to the process of self-monitoring, self-diagnosis of various aspects of menstrual health, hygiene and practices The participants of this study reported multiple practices as a self-testing practices including tracking menstrual flow, identifying irregularities, monitoring symptoms (such as pain or mood changes), and checking for signs of infections or other health concerns, use of pregnancy kit in case of missed periods among others.

The majority of the women with disabilities relied heavily on their body responses, such as gut feelings, head itches before menstruation, mood, and body temperature to track their menstrual cycle. Very few of the women were found to track menstruation using calendars because of limited awareness.

*As my menstrual cycle approaches, I notice that my scalp starts to itch, signalling the onset of my period. No, I do not use any calendar or mobile phone. I do not know how to use them. -* Woman with moderate disability.

In case of missed period, some of the participants shared using a pregnancy kit to check for potential pregnancy, while the majority didn't engage in this practice.

*Whenever I miss my period for 2 months, I use a pregnancy kit to test my pregnancy, This advice was given by FCHV.* - Woman with moderate disability.

The majority of the participants experienced pain and discomfort during menstruation, and it was often perceived as "normal" and "not requiring treatment". Some of the participants self-monitored the consistency and colour of the blood to know if the blood flow is normal or not, while some of them did not care about the consistency and flow of blood.

*Sometimes I observe my colour and the thickness of the menstrual blood to know if it is normal or not.* - Woman with mild physical disability.

**Self-awareness regarding menstrual practices.** This sub-theme discusses the extent of knowledge and awareness that participants have regarding their menstrual cycle, hygiene practices, and the sources from which they obtain relevant information. It also discusses self-help during menstrual practices and self-determination for menstrual hygiene, health, and well-being.

The majority of the participants were familiar with the literal meaning of menstruation and recognized the importance of maintaining menstrual hygiene to prevent vaginal infections. However, they had limited in-depth knowledge about topics such as the cause of menstruation, the physiology of menstruation, and many others.

*Menstruation is the natural process of monthly bleeding through the vagina in females. During menstruation, we should be consuming healthy food, should maintain hygiene, take bath regularly, and should keep our vagina clean.* - Woman with moderate physical disability

The majority of the participants perceived menstrual blood flow as impure and dirty blood and continued to separate themselves from family members during menstruation. This was due to deeply ingrained superstitious beliefs and taboos within their families, communities, and personal beliefs regarding menstruation. Some of the participants feared transferring disabilities to their unborn children with the regular intake of pain-relieving allopathic medicines. Some women feared that discontinuing traditional practices during menstruation could worsen their disability.

*I do not take any allopathic medicines to reduce my menstrual pain because, with an overdose of medicines, the chances of disability of a child in the womb will be high; therefore, I do not allow my daughter-in-law to take any medicines.* - Woman with severe physical disability.

In addition, some of the participants fully agreed on continuing the tradition of seclusion during menstruation, because they consider these days as a break from their regular gender roles.

*I agree with this tradition of staying separate during menstruation. In the name of tradition, at least we are getting a chance to take a rest. We do not need to cook food or carry loads of grass or water like every day. We become completely free to take proper rest. Though staying outside the house during the period is bad, however, staying within the home, outside the kitchen, is relaxing on those days.* - Woman with moderate physical disability.

On the contrary, some of the participants expressed feeling fear and discomfort when they had to sleep separately from other family members during menstruation, and they expressed a dislike for this tradition; however, they felt incapable of challenging the tradition.

*I don't like the tradition of sleeping separately away from the family members while menstruating, as I am always scared, and I need help with changing pads from my daughter; however, I can't even go against it.* - Woman with severe physical disability.

Regarding information and knowledge for making decisions about menstrual hygiene management, the majority of participants reported that their sources of information included peer groups, relatives, accessible mobile applications such as Pahunch and hami for SRHR [35,36] on menstrual hygiene management, training sessions, workshops organized by local organizations, and CBOs operating within their community. While young women with disabilities were found to be more active using mobile phones and digital information platforms to acquire disability inclusive knowledge and information, women who crossed 40 years of age were found to rely on radio broadcasts, FCHVs, and health professionals.

*I obtained information about menstrual hygiene management from the hospital, radio broadcasts, and occasionally through training sessions and workshops organized by various organizations. This information has helped me to combat difficulties such as back pain and stomach pain during my menstrual days. -* Woman with moderate physical disability.

Some younger participants aged between 18–20 years shared that they received information about menstruation from their parents and school, as menstruation-related topics are included in their curriculum. They stated that this knowledge has been helpful for them in practicing proper menstrual hygiene management, including changing pads, bathing, and the appropriate disposal of menstrual products. However, when asked whether the information provided in school was disability-inclusive, the majority of young participants responded otherwise.

*I have not received information about menstrual hygiene management from school from a disability-inclusive perspective. Whatever I have received (from school), I apply it in practice. -* Woman with moderate physical disability.

Similarly, participants shared mixed opinions regarding mental health issues such as fear and anxiety. The majority of women with severe disabilities reported feelings of inferiority because of their day-to-day dependency on family members. Women without access of proper knowledge, financial hardship, and lack of family support often experienced stigmatization. The combination of physical challenges, cultural norms, and social stigma contributed to feelings of embarrassment among women with physical disabilities concerning their menstrual health. On the contrary, women who had awareness, family support, and access to health facilities reported fewer mental health concerns.

*I wish I didn't have to depend on another person for my proper menstrual hygiene, as I don't feel good about it, and it makes me reluctant to talk about my further menstrual health-related problems as well. -* Woman with severe disability.

**Barriers and facilitators in the adoption of menstrual self-care practices**

**Existing gaps for the adoption of menstrual self-care practices.** The majority of the participants revealed the practical hardships and additional burdens due to their disability status and barriers in handwashing, changing pads, taking a shower, cleaning the genital organ, especially in harsh and cold weather conditions, due to limited availability of hot water, a cloth dryer, a water system, and other available technology.

*During my monthly period, I repeatedly face challenges washing my clothes, especially in cold weather. Handwashing, changing pads, and bathing become difficult due to the disability of my hands. Once, my hand swelled during snowfall, and I had to use hot water for bathing and laundry, which is again another hustle for me. -* Woman with moderate physical disability.

Financial hardship was another major barrier to the adoption of self-care practices, as women with physical disabilities could not afford health check-ups, medicines, menstrual products, and existing technologies. Apart from this, participants

also shared those economic challenges hindered their access to adequate nutrition and the purchase of menstrual products.

*I do not have money of my own since I do not work, so unless it is an emergency, I do not go to the hospital for any health issues. For my abdominal pain during menstruation, I drink hot water to reduce the pain.* - Woman with severe physical disability.

Likewise, women without limbs and those with severe physical disabilities often struggle with tasks like changing menstrual products, maintaining hygiene, and properly disposing of used pads. In addition, women with severe physical disabilities often had to rely on external help for assistance with menstrual hygiene management. This dependency has enforced a lack of autonomy, making it harder to practice self-care.

*I can't even think of managing menstrual hygiene on my own since I must fully rely on my family members. In such a situation, I rarely think about self-care practices for menstruation.* - Woman with severe disability.

Many of the participants shared that despite having menstrual discomfort, such as abdominal pain and backache during menstruation, they lack support from their husbands and family members. Therefore, the majority of the participants failed to implement self-treatment practices to lessen their menstrual discomfort.

*Sometimes I have a stomachache during my period. I cannot enter the kitchen to boil water for drinking and a hot bath, and my husband, as well as my family members, care less about me, thinking that menstrual pain is normal. Also, no one touches me currently, therefore I feel bad about it.* - Woman with moderate physical disability.

Even though the majority of the participants experienced pain and discomfort during menstruation, they perceived pain and discomfort as normal. Some of the participants had heavy blood flow and different colours of blood but did not reach out for consultations and treatment. Likewise, the majority of the participants were unaware of medical conditions like endometriosis, Polycystic ovary syndrome (PCOS), and Pelvic inflammatory diseases that cause painful menstruation.

*I think our whole body is bleeding while having menstruation, therefore we feel pain, as there could not be no other reason.* - Woman with moderate physical disability

The existing harmful social norms, values, and beliefs compounded by disabilities have been another barrier hindering the health and well-being of women with physical disabilities. the majority of the participants shared that during menstruation, women are not allowed to have certain dairy products, fruits, and foods like Yoghourt, ghee, milk, meat, pickles, among many others, that are important to have during menstruation. Upon probing, they stated that this restriction has been an entrenched cultural practice for a long time, and they are obliged to follow it.

*In the village, milk and curd are forbidden to the girls during their menstrual time; therefore, though I like to eat them, I do not eat rather practice my traditional culture.* - Woman with moderate physical disability.

**Facilitators for self-care menstrual practices.** The majority of the participants were aware of the importance of consuming a nutritious diet during menstruation. However, despite this awareness, some women with disabilities were unable to afford it due to financial constraints.

*During my period, I eat food that is available at home, like rice and lentils only. I know that we must eat a balanced diet during menstruation, especially, but we do not have the money to buy meat and eggs.* - Woman with severe physical disability.

Participants also emphasized that the information they received from various sources significantly contributed to improving their understanding of self-care practices during menstruation. This further helped them to challenge social taboos and stigma. Practical orientations and training from various community-based organizations targeting women with disabilities were particularly helpful in equipping them with menstrual hygiene management skills.

*My understanding regarding menstrual hygiene management after attending the training by our FCHV ma'am was far better than before. I knew the proper disposal of used pads. Before, I was too shy to dry my cloth pads in the sun, but now I do not feel shy at all.* - Woman with moderate disability.

A few participants also shared that they receive support from family members, particularly from their husbands and children, when they experience stomach aches and back pain during menstruation, which has helped them to self-manage the menstrual discomfort.

*My husband and daughter help me by bringing hot water and medicines, by asking what has happened, and by asking whether to go for a check-up.* - Woman with severe physical disability.

Additionally, digital platforms such as Facebook, mobile applications, newspapers, and magazines have been important platforms for women with physical disabilities with no access to community-based programs to acquire information and skills for menstrual practices for good health and well-being.

*Thanks to social media like Facebook as we can get lots of disability inclusive and other sexual and reproductive health-related information through it. I am an introvert, so it is not easy for me to open up; therefore, Facebook has been a great help in such situations.* - Woman with moderate physical disability.

## Discussion

This study presents a systematic description and understanding array of self-care practices, such as self-medication, self-examination, self-help, self-education, self-determination, self-efficacy, self-use, and self-treatment, practiced by women with physical disabilities specifically for self-management, self-testing, and self-awareness regarding menstrual health practices, which are the unique findings of this study [3]. In this study, women with physical disabilities living in rural areas face significant barriers such as inaccessible transportation, disability-exclusive roads, acceptance of menstrual taboos, harmful social norms and practices, limited information regarding menstrual hygiene and health management, and limited understanding of the importance of self-care for menstrual practices in accessing a self-care approach for the menstrual period. Similar social, cultural, physical, and economic barriers as presented in this study have been consistently reported by the studies conducted in rural areas of Nepal, Pakistan, and Ethiopia [37–41]. Another significant barrier to the adoption of a self-care approach in menstrual practice has been a lack of support from family members and communities due to cultural taboos around menstruation. This lack of support exacerbates the difficulties faced by women with disabilities in managing menstrual discomfort [42]. One of the major findings of this study has been self-treatment through home remedies due to the perceived harm of allopathic medicine on women, girls, and foetuses, which is a unique observation compared to other studies in Nepal, where women and girls consumed allopathic medicine to self-treat cramps and dysmenorrhoea [43]. The study conducted by Mason et. al suggests that dependence on

traditional remedies is prevalent in areas with limited access to healthcare services, suggesting the need for decentralization of services at the doorstep [44].

Limited knowledge and ignorance on menstruation have been a common observation in the studies conducted on menstrual hygiene, as highlighted by a systematic review conducted by Hennegan et.al. [45]. The study also reveals the same experiences faced by women with physical disabilities. However, unlike some studies [46] where women report significant health issues such as severe pain that requires medical attention during menstruation, our study revealed that the majority of the participants managed their discomfort on their own, and almost all the participants revealed not seeking health services even if there was discomfort during menstruation. This could be fatal sometimes, as the symptom such as pain, discomfort, and heavy blood flow could weaken women in general and could be a symptom of various diseases/infections such as dysmenorrhea, endometriosis, Polycystic Ovarian Disorder, reproductive tract infection or abnormal vaginal discharge, which could compromise health and well-being of women including women with disabilities [47–49]. The findings of this study suggest mental distress, particularly before (pre-menstrual syndrome and premenstrual dysphoric syndrome) and during menstruation, due to fear of seclusion and discrimination practiced during menstruation. Various studies have established the impact of mental health conditions such as anxiety, depression, and stress on the menstrual cycle and health [50,51]. This evidence suggests the need for knowledge, skills, and self-management techniques for mental health conditions throughout the menstrual cycle, which is often neglected by individuals, families, and institutions. Thus, prioritizing self-care in menstrual practices is essential to enhance women's health and well-being, prevent co-morbidities and complications related to menstrual health, and increase the autonomy of women and girls to make informed menstrual choices [52].

The trend of using menstrual products during menstruation was determined by the availability, accessibility, and affordability of the menstrual products. [45]. In this study, the women with physical disabilities using cloth pads highlighted the tediousness and difficulty in self-management of menstrual hygiene, particularly due to a lack of water, cold water during winter, and limited sunlight during winter. Additionally, the lack of information regarding various menstrual products and the low availability of a range of menstrual products for women to choose to have compromises hygiene and convenience for women. Public and private institutions need to identify, procure, inform, and offer diverse menstrual products for women with disabilities to optimize their menstrual health and hygiene [16].

The study significantly highlights the prevalence of existing taboos, harmful social norms, practices, and acceptance of such norms and practices, suggesting low access to information and education regarding menstrual health. This gap in understanding is consistent with findings from similar studies, which indicate that while basic menstrual hygiene knowledge is common, a deeper understanding of menstrual health is often lacking among women in various regions [53]. Thus, this study suggests that the provision and access to technologies and digital media could offer tailored educational resources, facilitate remote access to crucial and accurate menstrual health information, self-help skills to overcome physical and infrastructure barriers, and cater to the diverse needs of women with disabilities to practice and enhance autonomy for an optimum level of menstrual health management. Digital platforms provide customizable content such as videos with sign language interpretation for the hearing impaired and screen reader-compatible materials for the visually impaired [54,55]. Moreover, online forums and anonymous chat services create safe spaces where women can seek guidance and share experiences without fear of stigma [54,55].

Female Community Health Volunteers (FCHVs) in Nepal play a significant role in spreading awareness in the community and improving community knowledge on various health-related topics [56]. FCHVs are directly involved with females in the community, armed with basic knowledge of menstrual hygiene, they can create awareness among the targeted population, distribute sanitary pads, and dispel myths and superstitions [56]. The FCHVs can demonstrate steps to ensure self-care during menstruation by providing information about proper use of sanitary pads and their disposal, choice of materials, medication to alleviate pain if necessary, and information about the need for seeking treatment in case of infections [56]. Apart from this, while the FCHVs can be an important linkage between the community and health facilities, this

study recommends orienting and training rural health care providers for comprehensive and accurate menstrual health-related information, counselling, and decentralized quality menstrual health care [57].

Though the government of Nepal provides disability allowance that supports persons with disabilities financially [58,59], the participants of this study have significantly highlighted the economic constraints as an important barrier that can compromise the capacity to purchase menstrual products and compromise access to nutritious food, which is particularly important during menstruation when women may have increased nutritional needs [60]. This issue is especially pronounced in low-income communities where individuals frequently report running out of food by the end of the month and facing challenges in accessing affordable, nutritious options [61]. Apart from this, the water, sanitation, and hygiene (WASH) policy, infrastructure, including menstrual waste management and WASH programs in Nepal, are not disability inclusive, further compromising the rights of persons living with disability to menstrual hygiene and health [59].

Thus, this study recommends systemic changes fuelled by a comprehensive understanding and inclusion of self-care approaches in sexual and reproductive health policy and program, inclusion of social protection for women with disability, knowledge and access to choice in menstrual product, adequate utilization of digital SRH messaging platform, strengthened individual and community agency, and family support to address the multilayered and complex socio-ecological impact on health and well-being of women with physical disabilities [57–62]. Furthermore, this study recommends government policies, strategies, and programs, such as disability nutrition allowance/packages provided to extremely destitute children under the social security scheme, prioritize food assistance programs, promoting farmers' market, food security literacy, establishing nutrition rehabilitation centre, inclusion of women living with disabilities under health insurance program, to cover the economic viability of women with physical disabilities and to maintain adequate nutrition for optimum menstrual health management [58,63].

## Strengths and limitations of the study

One of the strengths of this study is the comprehensive documentation of current menstrual self-care practices and experiences among women with physical disabilities, which is crucial for health promotion, the prevention of sexual and reproductive morbidity, improved quality of life, and increased sexual and reproductive health autonomy. The study adopts an intersectional approach to understanding the relationship between gender, socio-economic context, disability, and self-care in menstruation, which constitutes another strength. To further mitigate researcher bias, all interviews were conducted and analysed by the well-trained principal investigator (PI) and co-principal investigator (Co-PI), who discussed and finalized the codes, sub-themes, and themes collaboratively with the research team to ensure neutrality and control personal biases and expectations. The study follows a strong design for the study, data collection processes, data analysis, ethical considerations, and reporting, which have ensured the study's credibility, dependability, and confirmability. However, the experiences and needs of women with other disabilities and those living in different geographical areas may differ and have not been investigated. Therefore, a nationally representative study on self-care in menstruation health management is needed to support the revision, formation, and/or implementation of national policies and programs related to menstrual health management.

## Conclusion

Women with physical disabilities often engage in self-care practices to manage menstruation. They soothe menstrual cramps mainly through home remedies and use cotton cloths as reusable pads to manage menstrual flow. In cases of missed periods, they independently use pregnancy test kits and mostly rely on physical symptoms to track their cycles. Observing blood flow allows them to self-identify conditions like menorrhagia or dysmenorrhea, reflecting their reliance on bodily awareness for menstrual health monitoring. However, due to disability, additional burdens such as the physical difficulty of washing reusable pads, feelings of reluctance stemming from day-to-day dependence on family members for personal tasks, and limited privacy or accessible facilities further complicate their ability to practice menstrual

self-care with dignity and independence. Self-awareness regarding menstrual myths, the knowledge and choice for varied menstrual products, and access to reliable and accessible information remain limited due to persistent social and cultural taboos. Strengthening self-care in menstruation requires improving access to inclusive and accessible digital platforms with menstrual health messaging, conducting community-based programs to promote self-care by raising awareness, debunking harmful myths, and sharing practical self-care strategies. Female Community Health Volunteers can play a critical role by providing tailored guidance, distributing sanitary products, demonstrating hygienic self-care practices during menstruation, and connecting women with disabilities to health services. Public and private sectors should ensure a range of menstrual products are available and offer counselling to support informed, autonomous choices. Ultimately, disability-inclusive menstrual health policies and multi-sectoral interventions that prioritize social protection and self-care should be integrated into national and local health systems through collaborative efforts across the health, nutrition, and agriculture sectors.

## Supporting information

**S1 File. Semi-structured interview guideline.**
(PDF)

**S2 File. Information sheet and consent sheet.**
(PDF)

## Acknowledgments

The research team would like to thank Center for Karnali Rural Promote and Society Development (CDS-PARK) Disable Rehabilitation and Rural Development Organization, Mugu, Blind Youth Association Nepal, Blind Women Association Nepal Surkhet Chapter, Chhayanath Rara municipality, and all the participants for their support and time.

## Author contributions

**Conceptualization:** Suyasha Adhikari, Pabitra Neupane, Anisha Shrestha, Amit Timilsina.

**Data curation:** Suyasha Adhikari, Pabitra Neupane, Anisha Shrestha, Manasi Sharma, Karishma Bhandari.

**Formal analysis:** Suyasha Adhikari, Pabitra Neupane, Anisha Shrestha, Manasi Sharma, Karishma Bhandari, Amit Timilsina.

**Investigation:** Suyasha Adhikari, Pabitra Neupane, Anisha Shrestha, Manasi Sharma, Karishma Bhandari, Amit Timilsina.

**Methodology:** Suyasha Adhikari, Pabitra Neupane, Amit Timilsina.

**Project administration:** Suyasha Adhikari, Pabitra Neupane, Anisha Shrestha, Manasi Sharma, Karishma Bhandari.

**Resources:** Suyasha Adhikari, Pabitra Neupane, Anisha Shrestha, Manasi Sharma.

**Software:** Suyasha Adhikari, Pabitra Neupane, Anisha Shrestha, Manasi Sharma.

**Supervision:** Amit Timilsina.

**Validation:** Suyasha Adhikari, Pabitra Neupane, Amit Timilsina.

**Visualization:** Suyasha Adhikari, Pabitra Neupane.

**Writing – original draft:** Suyasha Adhikari, Pabitra Neupane, Anisha Shrestha, Manasi Sharma, Karishma Bhandari, Amit Timilsina.

**Writing – review & editing:** Suyasha Adhikari, Pabitra Neupane, Anisha Shrestha, Manasi Sharma, Karishma Bhandari, Amit Timilsina.

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
