## [Decision Letter · Decision Letter 0]

14 Oct 2024

Dear Dr. Adhikari,

Thank you for submitting your manuscript to PLOS ONE. After careful consideration, we feel that it has merit but does not fully meet PLOS ONE’s publication criteria as it currently stands. Therefore, we invite you to submit a revised version of the manuscript that addresses the points raised during the review process.

**ACADEMIC EDITOR:**

We look forward to receiving your revised manuscript.

Kind regards,

Ranjit Kumar Dehury

Academic Editor

PLOS ONE

Journal Requirements:

Additional Editor Comments (if provided):

Dear authors,

You are advised to work very meticulously on the reviewers comments before reassessment for suitability of publication in the journal.

With regards,

Ranjit

Reviewers' comments:

Reviewer's Responses to Questions

**Comments to the Author**

1. Is the manuscript technically sound, and do the data support the conclusions?

Reviewer #1: Partly

Reviewer #2: Partly

2. Has the statistical analysis been performed appropriately and rigorously?

Reviewer #1: N/A

Reviewer #2: N/A

3. Have the authors made all data underlying the findings in their manuscript fully available?

Reviewer #1: Yes

Reviewer #2: Yes

4. Is the manuscript presented in an intelligible fashion and written in standard English?

Reviewer #1: Yes

Reviewer #2: Yes

Reviewer #1: The paper is written about a topic still emerging in current scientific discourse, so it is a valuable addition. However, some areas could be strengthened to ensure it reflects the existing evidence. For instance, the authors would benefit from drawing on existing data about disability and menstrual health in Nepal and other countries in their discussion. I have written detailed comments on each section in the attached document.

Reviewer #2: 1. I suggest instead of “disabled” and “women with physical disability”, the authors should use “differently abled” because "differently abled" reflects a more respectful, inclusive, and optimistic approach to a person with limited mobility, emphasizing their potential and unique capabilities. It aligns with the concept of "person-first language," which focuses on the person rather than their condition. This approach promotes the idea that an individual's identity is not defined by their physical conditions.

2. Why referencing is not done for Braun and Clarke in the references?

3. How 21 in-depth interviews and 1 FGD were determined as the sample?

4. How were the three organizations, namely DARRDO, BWAN, and BYAN, incorporated into the sampling process? Please explain their roles. BWAN and BYAN are associations for visually challenged persons; it is highly likely that your data mostly represent visually challenged persons while not the other types of physically challenged persons.

5. What are those severities and yellow & blue cards mentioned in Table 1? The manuscript does not explain how the type of disability can be understood using those terminologies.

6. Authors should mention how the identities of participants are anonymized. Who can access the data?

7. What are CBOs?

8. The full form of FCHV should come when it is first used in the manuscript, i.e. in the abstract.

9. “One of the participants.” before quotes is grammatically incorrect. Authors may write “One of the participants said” or “According to one of the participants;”

10. The authors confined the study to differently-abled married women, but they failed to capture the experiences that are unique to this group. Their result is general and not focused on the nature of the group studied. They can at least add two themes to the result, discussing the experiences that are unique to differently-abled married women regarding menstruation.

11. In the discussion section, various issues related to differently-abled persons’ challenges that are not found in the results section, such as mental health, diseases, and disorders.

12. In the discussion section, there is no need to write a literature review. You are supposed to substantiate and contradict your findings with the existing literature.

**Do you want your identity to be public for this peer review?** For information about this choice, including consent withdrawal, please see our Privacy Policy

Reviewer #1: No

Reviewer #2: **Yes: ** Imteyaz Ahmad

---

## [Author Response · Author response to Decision Letter 1]

10 Feb 2025

All the feedback provided by the reviewers has been uploaded.

---

## [Decision Letter · Decision Letter 1]

23 Mar 2025

Dear Dr. Adhikari,

Thank you for submitting your manuscript to PLOS ONE. After careful consideration, we feel that it has merit but does not fully meet PLOS ONE’s publication criteria as it currently stands. Therefore, we invite you to submit a revised version of the manuscript that addresses the points raised during the review process.

**ACADEMIC EDITOR:** The authors have to comply before reevaluation

We look forward to receiving your revised manuscript.

Kind regards,

Ranjit Kumar Dehury

Academic Editor

PLOS ONE

Additional Editor Comments:

The authors have to comply before reevaluation

Reviewers' comments:

Reviewer's Responses to Questions

**Comments to the Author**

Reviewer #1: (No Response)

Reviewer #2: (No Response)

2. Is the manuscript technically sound, and do the data support the conclusions?

Reviewer #1: Partly

Reviewer #2: Yes

3. Has the statistical analysis been performed appropriately and rigorously?

Reviewer #1: N/A

Reviewer #2: N/A

4. Have the authors made all data underlying the findings in their manuscript fully available?

Reviewer #1: Yes

Reviewer #2: Yes

5. Is the manuscript presented in an intelligible fashion and written in standard English?

Reviewer #1: Yes

Reviewer #2: Yes

Reviewer #1: Overall

This version is much improved, but it still needs strengthening for publication in an academic journal. In your revisions, use the checklist in Appendix 1 of this paper: https://pmc.ncbi.nlm.nih.gov/articles/PMC2987281/. It is very helpful for ensuring you are including all the relevant information in the paper and under which section it should go. The full paper explains how to present qualitative data for publication.

Abstract

• Methods: include your data analysis approach (eg thematic analysis)

• Result: Avoid using ‘etc’ in academic writing. Some results need more detail. You’re assuming the reader has prior knowledge of the topic, which mightn’t be true. If giving more detail this pushes you over your word count, reduce the number of results. E.g. Line 40 – can you give more details as the reader mightn’t understand the perceived cause and effect. Line 43-44: ‘feeling dependent and vulnerable’. There is a lot to unpack here. Line 48: ‘self-awareness’ or ‘knowledge’?

• Conclusion: don’t use acronyms (line 54). The conclusion should explain the implications of the findings and their relevance. You have included recommendations and these don’t clearly relate to the results.

Introduction

• Include a definition or explanation of ‘self-care’ by building on the WHO framework – eg self care is defined as xxx, which includes self-awareness, self-management… This is a considerable omission because the whole paper hangs on the practice self-care.

• Line 94: I’m not sure the disability cards ‘ensure’ inclusive development. Best use less definitive language, such as ‘To facilitate inclusive devt, the govt….’

• Line 108: needs a full stop and reference for the health issues

• When describing the situation in Nepal, can you include some data on Karnali so it builds the case for you doing your study there?

• Line 112-115. Without a definition of ‘self-care’ this sentence doesn’t make sense

• Line 116-118 – needs a reference

• Line 116-122: include your research objectives. You reference objectives in line 132 and you must include them explicity

Methods and materials

• Line 127 – reference Braun and Clarke and give a brief explanation of this approach

• Line 128 - Include information about Karnali province

• Participants and sampling:

o Include your definition of ‘physical disability.’ How did you determine this in your study (e.g. self-reported or from medical lists)?

o Line 130-133: ‘better explain their experiences according to the study’s objective’ introduces researcher bias as you could have excluded relevant participants. How did you attempt to manage that? How can you justify this approach?

o Why was ‘married women’ important?

o It would be helpful if you specifically state what your inclusion/exclusion criteria is

• Include a separate section on ethical considerations. You need to put a lot more information in about the ethical procedures you followed. This would include the informed consent process, which ethical board you sought and received ethical clearance from, where you conducted the interviews etc etc. Some of this is covered under ‘data collection and procedures’. You can look at published qualitative papers related to disability to see what should be included and the level of detail

• Reference the supporting documents were relevant (eg topic guides)

Results

• Line 199-202 and Table 2 should go under methods

• You use terms like ‘moderate’, ‘severe’ disability and severity of disability (line 262-263), but you have not explained what you mean by this and the criteria you used to define moderate in the methods. This introduces researcher bias. Either delete the reference to this throughout the results or explain the criterion in your methods.

• Line 226-227 – change to ‘their unborn children’ as that is what the participant is referring to, not living children

• Line 242 – define menstrual underwear. What are these, are they bought, are the reusable? Line 252 – you refer to menstrual panties, period panty. Use one term consistently throughout. In your introduction, include menstrual underwear as a product used in Nepal

• Line 245 – cloths, not clothes

• Line 281-285 – full stops are missing. Ensure you closely proof read your manuscript before submission

• In academic writing you always need to define terms used. Eg you refer to ‘proper disposal’; that must be explained as ‘proper’ can be subjective. Always use references when giving definitions. This doc might be helpful: https://www.unicef.org/documents/guidance-menstrual-health-and-hygiene

• Line 310: give an example to make this sentence clear

• Line 410: ‘some of the young’ – define the age range as you are only interviewing adults

Discussion

• Include a discussion on the transferability of research findings to other settings

• Line 515. Have a separate section on the study’s strengths and limitations

• Reflect on the influence of the researcher[s] on data, including a consideration of how the researcher(s) may have introduced bias to the results is included

Reviewer #2: 1. Abstract: Undefined acronym: FCHV

2. Results: The status of physical disability is mentioned with all the quotes, and age can also be mentioned for a better understanding of the background of the participants.

**Do you want your identity to be public for this peer review?** For information about this choice, including consent withdrawal, please see our Privacy Policy

Reviewer #1: No

Reviewer #2: **Yes: ** Imteyaz Ahmad

---

## [Author Response · Author response to Decision Letter 2]

22 May 2025

Response to Reviewer:

Reviewer #1:

Abstract

1. Methods: include your data analysis approach (eg, thematic analysis)

Answer: Thank you very much for your comment. We have incorporated it accordingly. Clean version line: 31 and 32.

2. Result: Avoid using ‘etc’ in academic writing. Some results need more detail. You’re assuming the reader has prior knowledge of the topic, which mightn’t be true. If giving more detail this pushes you over your word count, reduce the number of results. E.g. Line 40 – can you give more details as the reader mightn’t understand the perceived cause and effect. Line 43-44: ‘feeling dependent and vulnerable’. There is a lot to unpack here. Line 48: ‘self-awareness’ or ‘knowledge’?

Answer: Thank you for the comment and feedback. This is the abstract section with limited wordcount, so we have summarized the findings in the abstract section, while the detailed explanation has been provided in the Results section of the manuscript (not abstract). However, we have explained within the given word count for better clarity. The terminology 'self-awareness' is the standard terminology proposed by the WHO self-care framework, which we have adapted in this study. Clean version: Line: 35 to 44.

3. Conclusion: Don’t use acronyms (line 54). The conclusion should explain the implications of the findings and their relevance. You have included recommendations, and these don’t clearly relate to the results.

Answer: Thank you for the comment. We have incorporated it accordingly. Clean version Line 47 to 52.

Introduction

4. Include a definition or explanation of ‘self-care’ by building on the WHO framework – eg self care is defined as xxx, which includes self-awareness, self-management… This is a considerable omission because the whole paper hangs on the practice self-care.

Answer: Thank you for highlighting this. we have incorporated the definition of self-care on WHO framework in the line 58 to 63 of clean version.

5. Line 94: I’m not sure the disability cards ensure’ inclusive development. Best use less definitive language, such as ‘To facilitate inclusive devt, the govt….’

Answer: Thank you for the comment. We have adjusted the sentence accordingly in the line 114 of the clean version.

6. Line 108: needs a full stop and a reference for the health issues

Answer: Thank you for highlighting this. I have incorporated it accordingly in the line 129 of the clean version.

7. When describing the situation in Nepal, can you include some data on Karnali so it builds the case for you doing your study there?

Answer: Thank you very much for the insights. We have incorporated it from line 107 to 113 in the clean version of manuscript.

8. Line 112-115. Without a definition of ‘self-care’ this sentence doesn’t make sense

Answer: Thank you for pinpointing this. We have included the definition of self-care from the line 56 to 59.

9. Line 116-118 – needs a reference

Answer: Thank you very much for the comment. We have included the references accordingly in the line 136.

10. Line 116-122: include your research objectives. You reference objectives in line 132 and you must include them explicitly

Answer: Thank you for highlighting this. I have included the objectives in the clean version of the manuscript from line 136 to 140.

Methods and materials

11. Line 127 – reference Braun and Clarke, and give a brief explanation of this approach

Answer: Thank you very much for the comment. We have kept the reference and have provided a brief explanation of this approach. Line 143 to 150 in the clean version of manuscript.

12. Line 128 - Include information about Karnali province

Answer: Thank you very much for the comment. It has been incorporated accordingly from the line 105 to 111 of the clean version.

Participants and sampling:

13. Include your definition of ‘physical disability.’ How did you determine this in your study (e.g. self-reported or from medical lists)?

Answer: we have included the definition of physical disability in the introduction from the line 93 to 96 in the clean version of manuscript. Physical disability was chosen as it has a high prevalence compared to other disabilities (according to the Census of Nepal 2021). Three organizations of persons with disabilities (OPD) (Disable Rehabilitation & Rural Development Organization (DARRDO) Nepal, Blind Women Association Nepal (BWAN) Surkhet Chapter, and Blind Youth Association Nepal (BYAN) Surkhet Chapter) facilitate the determination of potential participants. This has been described in Line 153 to 159 of the clean version of manuscript.

14.Line 130-133: ‘better explain their experiences according to the study’s objective’ introduces researcher bias as you could have excluded relevant participants. How did you attempt to manage that? How can you justify this approach?

Answer: Thank you very much for the comment. I have included the objective of the study, removing the researcher bias. Only individuals who voluntarily agreed to participate and provided informed consent to share their experiences were listed. Women with physical disabilities under 18 years of age were excluded, as marriage before 18 years is illegal in Nepal. Additionally, any participants who did not provide informed consent were excluded. Based on these criteria, the principal investigator finalized the list of participants. This has been incorporated from the line 161 to 165 of the clean version of manuscript.

15. Why was ‘married women’ important?

Answer: According to a 2022 report by The Rising Nepal, approximately 52% of child marriages occur in Karnali Province of Nepal. Thus, women of reproductive age (15-49 years) are mostly married in Karnali province thus, married women were taken as the study population.

16. It would be helpful if you specifically state what your inclusion/exclusion criteria is?

Answer: Thank you very much for your comment. We have specifically stated the inclusion and exclusion criteria as per your suggestion. This has been well explained in the Participants and Sampling subheading from the line 152 to 165 in the clean version of the manuscript.

17. Include a separate section on ethical considerations. You need to put a lot more information in about the ethical procedures you followed. This would include the informed consent process, which ethical board you sought and received ethical clearance from, where you conducted the interviews etc etc. Some of this is covered under ‘data collection and procedures’. You can look at published qualitative papers related to disability to see what should be included and the level of detail

Answer: Thank you very much for the insight. We have added a section on ethical consideration as suggested by the reviewer in the line 209 to 219 in the clean version of Manuscript.

18. Reference the supporting documents were relevant (eg topic guides)

Answer: Since it was made under the description made by Barun and Clarke and SAGE Handb Qual Res Des. 2022;(April):290–306, this has been referenced. Also, the interview guide and consent form have been submitted as supporting documents in the submission portal.

Results

19. Line 199-202 and Table 2 should go under methods

Answer: The codebook and explanation of sub-themes and themes are results/findings of the study, rather than the method. So, we have kept it in the results section only. Thank you for your feedback.

20. You use terms like ‘moderate’, ‘severe’ disability and severity of disability (line 262-263), but you have not explained what you mean by this and the criteria you used to define moderate in the methods. This introduces researcher bias. Either delete the reference to this throughout the results or explain the criterion in your methods.

Answer: Thank you very much for your comment. All the terms are described under the introduction section to remove any confusion or ambiguity. Line: 96 to 103 of the clean version of Manuscript.

21. Line 226-227 – change to ‘their unborn children’ as that is what the participant is referring to, not living children

Answer: Thank you very much for pinpointing it. We have corrected accordingly. Line 396 of the clean version of manuscript.

22. Line 242 – define menstrual underwear. What are these, are they bought, are the reusable? Line 252 – you refer to menstrual panties, period panty. Use one term consistently throughout. In your introduction, include menstrual underwear as a product used in Nepal

Answer: Thank you for the comment. As per your feedback, I have made the consistent terminology as “menstrual underwear”. Menstrual underwear or period underwear are absorbent underpants that have multiple layers of microfiber polyester. They look like regular underwear, but they’re designed to keep moisture away from the skin as they soak up menstrual blood. The fabric in period underwear contains a moisture-wicking fabric made up of thousands of small filaments. These fibers trap blood or other liquid to keep it from leaking onto the clothes. The outer layer usually includes nylon and Lycra, and is then finished with a liquid-repellent film. ( https://www.webmd.com/women/period-underwear). Not only in Nepal but however, over the world, menstrual underwear is one of the menstrual product that is worn in during the mensturation to absorb the menstrual blood.

23. Line 245 – cloths, not clothes

Answer: Thank you very much for the comment. We have included it accordingly in the line 278 of the clean version of the manuscript.

24. Line 281-285 – full stops are missing. Ensure you closely proofread your manuscript before submission

Answer: Thank you very much for highlighting this. All the grammars in the manuscript have been thoroughly checked, proofread and ensured.

25. In academic writing, you always need to define terms used. Eg you refer to ‘proper disposal’; that must be explained as ‘proper’ can be subjective. Always use references when giving definitions. This doc might be helpful: https://www.unicef.org/documents/guidance-menstrual-health-and-hygiene

Answer:Thank you for your feedback and this has been well noted.

26. Line 310: Give an example to make this sentence clear.

Answer: Thank you very much for the comment. We have provided specific example on this. Line 351 to 366 of the clean version of manuscript.

27. Line 410: ‘some of the young’ – define the age range as you are only interviewing adults

Answer: Thank you very much for the comment, we have included the age range

(18-20)years. The comparison on the age range led to use the word young despite being all of them adult. This has been incorporated in the line 436 of the clean version of manuscript.

Discussion

28. Include a discussion on the transferability of research findings to other settings

Answer: Thank you very much for the comment. This has been explained under the section strengths and limitation. Line: 646 to 661 of the clean version of manuscript.

29. Line 515. Have a separate section on the study’s strengths and limitations

Answer: Thank you very much for the comment. This has been explained under the section strengths and limitation. Line: 646 to 661 of the clean version of the manuscript.

30.Reflect on the influence of the researcher[s] on data, including a consideration of how the researcher(s) may have introduced bias to the results is included

Answer: Thank you very much for the comment. This has been explained under the section strengths and limitation. Line: 646 to 661 of the clean version of the manuscript.

Reviewer #2:

1. Abstract: Undefined acronym: FCHV

Answer: Thank you for highlighting this. We have replaced the word with community volunteers in the clean version of the manuscript. Line: 46 of the clean version of manuscript.

2. Results: The status of physical disability is mentioned with all the quotes, and age can also be mentioned for a better understanding of the background of the participants.

Answer: To maintain anonymity and not provide information that could breach the anonymity, the participant's age has not been included. This approach was also followed during the first-round review process, in accordance with the given instructions.

---

## [Editor Report · Decision Letter 2]

14 Jul 2025

Dear Dr. Adhikari,

Thank you for submitting your manuscript to PLOS ONE. After careful consideration, we feel that it has merit but does not fully meet PLOS ONE’s publication criteria as it currently stands. Therefore, we invite you to submit a revised version of the manuscript that addresses the points raised during the review process.

We look forward to receiving your revised manuscript.

Kind regards,

Seifadin Ahmed Shallo, MPH

Academic Editor

PLOS ONE

**Journal Requirements:**

**Additional Editor Comments:**

Dear authors,

I appreciate your efforts in addressing then comments given you so far. Yet, the following issues also need to be addressed. 

1. How did you analyze your data (Data analysis method/software vs manual )

2. Is it monthly or life time history you took. The women may use sanitary pad sometimes and rag clothes on another time. The recent one she may used can affect your finding? (ups and downs in life )

3. Which type of physical disabilities did you consider in the study and why????

4. Why only married women? Since all women who started menstruations can be parts of the study.

5. Why only 1 FGD? What was your justification and how did you able to determine the idea saturation with single FGD????

6. Its better to indicate what severe, moderate, and mild disability mean in method section and what is your reference to classify them in such category??

7. If severity matters, did you consider including them from different severity status??

8. The way dta collected was bit confusing? Make it clear.

9. You have cited similar information with different source line…145 and 225. (refe 28 vs 31)

10. Did you use the deductive or inductive approach while you generate the “Themes’’? and why?

11. Is your finding different from what is common among general population? Especially in terms of menstrual hygiene materials utilized, awareness level, and in terms of pain relieving approaches?

12. The recommendation should also some thing inline with your finding and applicable (should be to bring tangible change if applied) 

---

## [Author Response · Author response to Decision Letter 3]

21 Oct 2025

1.How did you analyze your data (Data analysis method/software vs manual )

Answer: Thank you for your comment. The transcriptions and translations of the data were conducted verbatim, manually. For the coding of the translated data, inductive coding was conducted using Dedoose as data management software. This has been clearly incorporated in lines 215 to 230 of the clean manuscript

2.Is it a monthly or lifetime history you took. The women may use sanitary pad sometimes and rag clothes on another time. The recent one she may used can affect your finding? (ups and downs in life )

Answer: We collected participants’ lifetime histories documenting the sanitation practices and pad use, and the reasons for it. The participants in this study used either a sanitary pad or used clothes as a pad. Depending on the preference of the pad, the research focused on sanitation practices, which have been documented in the result section, lines 271 to 306 of the clean manuscript.

3.Which type of physical disabilities did you consider in the study and why????

Answer The participants in the study included individuals with varying degrees of severity of disability, profound (persons who face difficulty to perform day-to day activities even with continuous support of others), severe disability (persons who requires others support continuously to perform personal activities and involve in social activities), moderate disability (persons who can regularly participate in daily and social activities if physical, environmental, educational barrier is ended and mild disability (persons who can regularly participate in daily and social activities if physical and environmental barriers is dismantled (Line:97 to 103) as they are available physical disabilities of the clean manuscript.

Why only married women? Since all women who started menstruations can be parts of the study.

Answer: Choosing married women as participants helps to understand the experiences of menstruation from adolescence to adulthood, thus providing a comprehensive understanding of issues. The age of married women in this study ranges from 18-, thus gathering evidence from diverse women. Furthermore, compared to unmarried women with physical disabilities, married women with physical disabilities find it convenient to talk culturally about tabooed topics such as menstrual hygiene and can provide in-depth information from existing social norms, practices, and challenges for menstrual health management

4.Why only 1 FGD? What was your justification and how did you able to determine the idea saturation with a single FGD????

Answer: The study adopts IDI and FGD as data collection methods. We conducted 21 in-depth interviews. The FGD with 10 participants further helped to triangulate and validate the findings from the IDIs and helped to add information to reach the saturation. We adopted the definition of saturation from Fusch et. al- Are we there yet, as stated in lines 205 to 209 of the method section of the clean manuscript.

5.Its better to indicate what severe, moderate, and mild disability mean in method section and what is your reference to classify them in such category??

Answer: Thank you so much for your feedback. Thus has been discussed in the line 164 to 173 of the clean manuscript.

6.If severity matters, did you consider including them from different severity status??

Answer: This study has included different severity status- severe and moderate disability (2 out of 4 category) as the participants from these two categories has differential experience, as the objective of the study was to understand how the physical disability influences the menstrual hygiene experiences and practices. Even though initially, women with mild physical disability was included, they were excluded from the study, in consultation with partner organization, as there was no differential experience or impact to women without disability. This has been explained in the line 149-151 of the clean manuscript.

7.The way dta collected was bit confusing? Make it clear.

Answer: Thank you for your feedback, necessary edits has been made to make it clear from line .176 to 202.

8.You have cited similar information with different source line…145 and 225. (refe 28 vs 31)

Answer: Thank you for flagging this. We have edited the reference section accordingly.

9.Did you use the deductive or inductive approach while you generate the “Themes’’? and why?

Answer: We used an inductive approach for coding and aggregating codes to form sub-themes. The sub-themes with similar characteristics were aggregated to form themes as explained in the line: 227-230 of the clean manuscript.

10.Is your finding different from what is common among general population? Especially in terms of menstrual hygiene materials utilized, awareness level, and in terms of pain relieving approaches?

Answer: There are few observations such as: self-testing which is body symptoms, self-management of menstrual pain particularly using home remedies, and mental distress, particularly before (pre-menstrual syndrome and premenstrual dysphoric syndrome) and during menstruation due to fear of seclusion and discrimination practiced during menstruation. Similarly, women with physical disabilities using cloth pads highlighted the tediousness and difficulty in self-management of menstrual hygiene due to a lack of water and sunlight during winter, limited information and choice of menstrual product, limited access to menstrual health related information are some of the differential impact among women with physical disabilities compared to general population. This paper presents existing ways of self-care in menstrual health management and proposes way forward on how strengthening self-care could help enhance good health and well-being of women. This has been well explained in detail in result and discussion section of the manuscript.

11.The recommendation should also some thing inline with your finding and applicable (should be to bring tangible change if applied)

Answer: Thank you for your suggestion. We have proposed recommendations to strengthen self-care in menstrual health management for the overall health and well-being of women and girls, aligning with the findings in the discussion section from line 594 to 638 of the clean manuscript.

---

## [Editor Report · Decision Letter 3]

19 Nov 2025

Understanding the menstrual health self-care practices and experiences among women with physical disabilities in rural Nepal: A qualitative study 

PONE-D-24-31534R3

Dear Dr. Adhikari,

We’re pleased to inform you that your manuscript has been judged scientifically suitable for publication and will be formally accepted for publication once it meets all outstanding technical requirements.

Kind regards,

Seifadin Ahmed Shallo, MPH

Academic Editor

PLOS ONE

---

## [Editor Report · Acceptance letter]

1 Dec 2025

PONE-D-24-31534R3

PLOS ONE

Dear Dr. Adhikari,

I'm pleased to inform you that your manuscript has been deemed suitable for publication in PLOS ONE. Congratulations! Your manuscript is now being handed over to our production team.

Kind regards,

on behalf of

Prof. Seifadin Ahmed Shallo

Academic Editor

PLOS ONE